# Cerebral blood flow changes during palpation of external airway structures in healthy volunteers

Paul S. Basel[1], Michael D. April[2], Allyson A. Arana[3], Jessie Renee D. Fernandez[3], Steven G. Schauer[2,3,4]*

**1** Joint Base Elmendorf-Richardson, Anchorage, Alaska, United States of America, **2** Brooke Army Medical Center, JBSA Fort Sam Houston, San Antonio, Texas, United States of America, **3** 59[th] Medical Wing, JBSA Lackland, San Antonio, Texas, United States of America, **4** US Army Institute of Surgical Research, JBSA Fort Sam Houston, San Antonio, Texas, United States of America

* steven.g.schauer.mil@mail.mil

## Abstract

**Data Availability Statement:** All relevant data are within the paper and its Supporting Information files.

### Introduction

Previous studies demonstrate increased intracranial pressure (ICP) during direct laryngoscopy in patients with traumatic brain injury (TBI). Worse outcomes in TBI have been associated with increased ICP. It remains unclear if the same effect occurs during cricothyrotomy. We evaluated changes in cerebral blood flow and hemodynamic changes that occurred during preparation for cricothyrotomy in healthy volunteers.

### Methods

An emergency medicine trainee performed routine anatomical procedural palpation with simultaneous transcranial doppler (TCD) measurements of cerebral blood flow velocities (CBFV) from bilateral middle cerebral arteries (MCAs). Mean arterial pressure (MAP) and heart rate (HR) were recorded throughout event. Our primary outcome was changes in pulsatility index (PI) and CBFV by TCD during palpation. TCD measurements were used as a surrogate for ICP.

### Results

We enrolled 20 healthy volunteers for this study. No significant differences were found in pulsatility index [Right MCA -0.02 (95% confidence interval, -0.09 to 0.06), left MCA -0.02 (95% confidence interval, -0.011 to 0.07)] or mean CBFV [right MCA -0.70 mm/s (95% confidence interval, -10.15 to 8.75) left MCA -1.20 mm/s (95% confidence interval, -10.68 to 8.28)] during palpation. No significant change in HR was found [-1.1 bpm ((95% confidence interval, -2.4 to 0.1)]. A change in MAP was observed [1.3 mmHg (95% confidence interval, -0.1 to 2.4)].

**Funding:** Our study was funded by the San Antonio Uniformed Services Health Education Consortium as part of the graduate medical education mission.

**Competing interests:** We have no conflicts to report.

## Conclusions

In healthy individuals, no clinically significant change in cerebral blood flow velocities, ICP, or change heart rate was observed during palpation for cricothyrotomy.

## Introduction

Traumatic brain injury (TBI) is the most common cause of death for individuals under age 45 in America according to the Center for Disease Control and Prevention. [1] In recent years, TBI has become an increasingly prevalent problem facing the United States (US) military. Rates of TBI among service members doubled from 2000 to 2011 due to injury patterns experienced during Operation Enduring Freedom and Operation Iraqi Freedom. [2]

Key to the management of severe TBI is prevention of secondary injury. This includes preventing hypoxia and increased intracranial pressure (ICP) which are both independent predictors of poor outcomes in severe TBI. [3–6] Many TBI patients require intubation for the purposes of preventing hypoxia and ensuring airway protection [7–9]. Unfortunately, airway management by direct laryngoscopy can predispose patients to ICP spikes due to manipulation of the oropharynx. [10,11] It is unclear whether the airway manipulation required by alternative airway management techniques causes similar rises in ICP.

Cricothyrotomy is an emergent procedure to cannulate the trachea and is used early in the airway management algorithm under Tactical Combat Casualty Care guidelines [12]. Moreover, most military combat medics are not trained in endotracheal intubation so cricothyrotomy is the primary airway intervention in the setting of anatomical disruptions.

Another potential source of secondary injury in TBI patients aside from hypoxia is increased intracranial pressure. Data suggests an association between increased intracranial pressure and worse outcomes in the setting of TBI. [13] Changes in cerebral blood flow (CBF) can have deleterious effects in TBI patients, even if transient. [4] Previous literature demonstrated that airway manipulation via direct laryngoscopy causes changes in cardiovascular hemodynamics and intracranial pressure which can affect cerebral blood flow [14–16]. It is unclear at this time how airway management via cricothyrotomy affects cerebral blood flow and hemodynamics. In this study we sought to determine the change in CBF during the palpation of neck landmarks in preparation for cricothyrotomy.

## Materials and methods

We conducted a prospective observational study of volunteer subjects. The study setting was an patient room in an urban, tertiary-care hospital emergency department (ED). The Brooke Army Medical Center (Regional Health Command-Central) institutional review board reviewed and approved protocol C.2016.089.

We recruited healthy subjects 18 years of age and older for participation. We excluded subjects with a history of increased ICP, abnormal skull or cerebral anatomy precluding the ability to obtain transcranial Doppler (TCD) measurements, and abnormal or altered neck anatomy precluding the ability to palpate the neck as require for the study (i.e. goiter, thyroid surgery, etc). We did not compensate subjects for participation. We provided all subjects with information regarding the study procedures and consent was obtained.

We placed all subjects in a supine position on a stretcher. We next used a measuring tape to measure and record neck circumference at the level of the cricothyroid membrane. An

emergency medicine resident physician (PB) palpated the larynx for stabilization in their routine manner in preparation for a cricothyrotomy procedure. The physician maintained this palpation for 30 seconds at which time we repeated physiologic measurements. The physician then stopped palpation and the subject remained in the supine position for one minute after which we took final physiologic measurements.

Demographic information was obtained prior initiating study procedures. Physiologic measurements taken throughout the study included mean arterial pressure (MAP), pulse oximetry, and cerebral blood flow velocity (CBFV). We measured blood pressure (BP) and pulse oximetry through use of a Nexfin device utilizing finger plethysmography (BMEye, Amsterdam, The Netherlands). We utilized transcranial Doppler (TCD) measurements as a non-invasive method of monitoring CBFV. TCD is currently the only modality that provides a reasonable measurement of CBFV without the use of invasive measurement tools. [17] It is very sensitive for increased ICP and measurements correlate well with invasively-determined pressures as measured by ventriculostomy. [18–21] We measured CBFV using the Spencer Technology ST3 Digital Transcranial Doppler System model PMD150 (Spencer Technologies, Redmond, WA). Specifically, a trained TCD technician took CBFV measurements from each subject's bilateral middle cerebral arteries (MCA). We obtained these measurements prior to initiation of study procedures with the patient in the supine position, after 30 second of physician airway landmark palpation, and finally one minute after completion of all palpation maneuvers. All data was recorded in real time using hard copy data collection forms. The study team then converted all measurements into a format for storage in an Excel Database (version 14, Microsoft, Redmond, WA).

The primary outcome was change in MCA pulsatility index during palpation for cricothyrotomy. We calculated pulsatility index according to the following equation:

$$PI = \frac{FVs - FVd}{FVm} \tag{1}$$

Where FVs = Systolic Flow Velocity; FVd = Diastolic Flow Velocity and FVm = Mean Flow Velocity.

Secondary outcomes include changes in mean CBFV, changes in mean arterial pressure, and changes in heart rate.

## Results

We approached 20 individuals for study inclusions. All 20 persons were eligible for and agreed to enroll in the study. All 20 subjects completed all study procedures. The mean age of subjects was 32 years and predominantly male (80%, Table 1).

We observed no significant differences in mean pulsatility index or CBFV of the right and left MCAs between baseline, palpation, and post palpation measurements (Table 2, Fig 1). Physiologic measurements were similarly comparable across the three time measurements

**Table 1. Patient demographics.**

|  | Mean | Range |
|---|---|---|
| Age (yrs.) | 32 | 27–42 |
| Male (%) | 80 | |
| Neck circumference (cm) | 38 | 30–42 |
| Weight (kg) | 77 | 50–98 |
| Height (cm) | 176 | 160–188 |

**Table 2. Pulsatility index and cerebral blood flow velocity measurements.**

| | Mean (SD) | | | | |
|---|---|---|---|---|---|
| | Baseline | Start palp | End palp | Baseline vs. start palp | Baseline vs. end palp |
| **PI–Right** | 0.77 (0.12) | 0.78 (0.08) | 0.75 (0.09) | 0.02 (-0.06, 0.09) | -0.02 (-0.09, 0.06) |
| **PI–Left** | 0.77 (0.13) | 0.78 (0.12) | 0.75 (0.09) | 0.01 (-0.01, 0.01) | -0.02 (-0.11, 0.07) |
| **CBFV–Right** | 59.00 (11.87) | 58.85 (12.59) | 58.30 (11.86) | -0.15 (-9.60, 9.30) | -0.70 (-10.15, 8.75) |
| **CBFV–Left** | 58.40 (12.16) | 58.00 (12.72) | 57.20 (11.56) | -0.40 (-9.88, 9.08) | -1.20 (-10.68, 8.28) |

(Table 3). While there was a rise in mean arterial pressure (MAP) which reached statistical significance, the effect size difference is unlikely to be clinically significant (1.26 mm Hg).

## Discussion

Based on TCCC guidelines, cricothyrotomy is the primary invasive method for securing an airway in the prehospital combat environment. Many patients requiring airway management in this setting will have concomitant TBI. Avoiding rises in ICP is critical to optimize

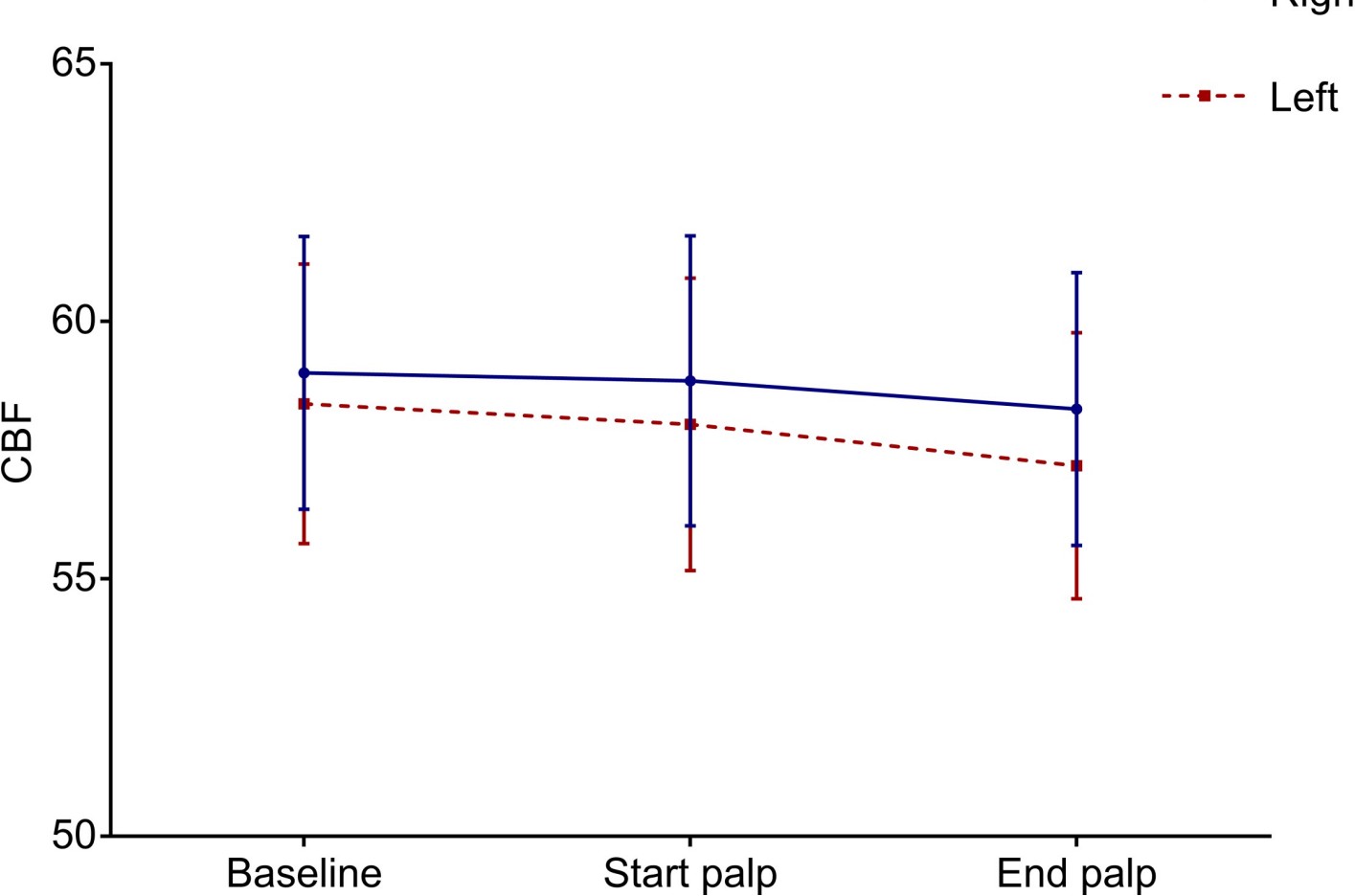

**Fig 1. Cerebral blood flow velocity.** The horizontal axis separates the three separate times at which we took measurements: pre-palpation (baseline), after 30 seconds of initial palpation (start palpation), and one minute after completion of palpation (end palpation). The vertical axis represents cerebral blood flow velocity (cm/s). The blue solid line represents mean values for all right MCA measurements. The red dashed line represents mean values for all left MCA measurements. The vertical bars represent the 95% confidence intervals for each of the mean values.

**Table 3. Physiologic measurements.**

|  | Baseline | Palpation | Difference (95% CI) |
|---|---|---|---|
| **HR (beats per minute)** | 58.3 (6.2) | 57.2 (5.7) | -1.1 (-2.4, 0.1) |
| **MAP (mm Hg)** | 90.5 (7.3) | 91.8 (8.4) | 1.3 (0.1, 2.4) |

outcomes among these patients given the importance of maintaining cerebral blood flow. Prior to this study, the impact of the palpation of neck landmarks in preparation for the cricothyrotomy procedure on cerebral blood flow was unclear. To our knowledge, this is the first study to evaluate changes in cerebral blood flow during cricothyrotomy. Using healthy volunteers that mimic the typical combat casualty population, we found no evidence of increased intracranial pressure during palpation for cricothyrotomy. Methods to reduce the ICP changes that occur during cricothyrotomy are needed–our study suggests that the palpatory preparation stage is not a procedural task that requires changes or interventions to avoid ICP spikes.

Previous studies have evaluated airway manipulation during standard laryngoscopy. These studies showed significant increases in ICP as well as systemic blood pressure and heart rate with laryngoscopy and tube placement. [10,11,14,15,22–26] However, we did not find that such external manipulation caused the same effects. It may be that manipulation of the epiglottis and vallecula leads to more ICP alterations rather than palpation of external airway structures.

Our results do not support any changes to current guidelines for performance of the cricothyrotomy procedure or utilization of this procedure by prehospital providers in the deployed environment. Our data does suggest that further research is needed to find the procedural task that can be best targeted to develop methods to blunt ICP effects from the procedure. We recommend further studies using a large animal model with similar anatomy to the human airway. Such a study is not feasible in humans given the invasiveness of the procedure, as such, we limited our study to external measurements only.

Our study has several limitations. First, we enrolled subjects that are young, healthy and without significant injury, specifically TBI. We do not know how the presence of a TBI would have altered our findings. Second, we evaluated only the palpation portion of cricothyrotomy. We are unable to assess which invasive steps may lead to ICP spikes, including internal airway structure palpation. Third, we used non-invasive measurements to estimate ICP. While several studies have shown correlation between CBFV as measured by TCD and ICP it is possible that invasive monitoring would reveal more significant changes [21,27,28]. Fourth, a single provider performed all palpations. It is possible that landmark palpation by other providers would have yielded greater impacts on our measurements if they used a different preparatory palpation technique, more force, or more directional changes for example. Fifth, due to ethical issues, we had to use healthy individuals that were not experiencing injuries or physiology that would frequently be present after polytrauma (e.g. hypoxia, TBI, etc.). As such, our results should be considered informative for future research and hypothesis generating.

## Conclusion

In healthy individuals, no clinically significant changes in cerebral blood flow velocities, ICP, or changes heart rate were observed during palpation for cricothyrotomy.

## Supporting information

**S1 Data.**
(XLSX)

## Acknowledgments

**Disclaimer:** The view(s) expressed herein are those of the author(s) and do not reflect the official policy or position of Brooke Army Medical Center, the U.S. Army Medical Department, the U.S. Army Office of the Surgeon General, the Department of the Army, the Department of the Air Force and Department of Defense or the U.S. Government.

## Author Contributions

**Conceptualization:** Michael D. April, Steven G. Schauer.

**Data curation:** Paul S. Basel, Jessie Renee D. Fernandez.

**Formal analysis:** Paul S. Basel, Steven G. Schauer.

**Investigation:** Paul S. Basel, Steven G. Schauer.

**Methodology:** Allyson A. Arana, Steven G. Schauer.

**Project administration:** Paul S. Basel.

**Supervision:** Michael D. April, Steven G. Schauer.

**Validation:** Michael D. April, Allyson A. Arana.

**Writing – original draft:** Paul S. Basel.

**Writing – review & editing:** Michael D. April, Allyson A. Arana, Jessie Renee D. Fernandez, Steven G. Schauer.

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
