## [Decision Letter · Decision Letter 0]

7 May 2020

PONE-D-20-03670

Cerebral Blood Flow During Palpation for Cricothyrotomy

PLOS ONE

Dear Dr. Steven G. Schauer, 

Thank you for submitting your manuscript to PLOS ONE. After careful consideration, we feel that it has merit but does not fully meet PLOS ONE’s publication criteria as it currently stands. Therefore, we invite you to submit a revised version of the manuscript that addresses the points raised during the review process.

We would appreciate receiving your revised manuscript by May 25, 2020. To enhance the reproducibility of your results, we recommend that if applicable you deposit your laboratory protocols in protocols.io, where a protocol can be assigned its own identifier (DOI) such that it can be cited independently in the future. For instructions see: http://journals.plos.org/plosone/s/submission-guidelines#loc-laboratory-protocols

We look forward to receiving your revised manuscript.

Kind regards,

Alon Harris

Academic Editor

PLOS ONE

Journal requirements

1.When submitting your revision, we need you to address these additional requirements.

Reviewers' comments:

Reviewer's Responses to Questions

**Comments to the Author**

1. Is the manuscript technically sound, and do the data support the conclusions?

Reviewer #1: Partly

Reviewer #2: Partly

2. Has the statistical analysis been performed appropriately and rigorously? 

Reviewer #1: No

Reviewer #2: Yes

3. Have the authors made all data underlying the findings in their manuscript fully available?

Reviewer #1: Yes

Reviewer #2: Yes

4. Is the manuscript presented in an intelligible fashion and written in standard English?

Reviewer #1: Yes

Reviewer #2: Yes

5. Review Comments to the Author

Reviewer #1: It is a well written article. Patients requiring cricothyrotomy are usually hypoxic and this itself can raise the ICP and cause changes in the cerebral blood flow. Examining normal patients and palpating their cricothyroid has not shown a change in the cerebral blood PI and CBFV in your study. You have not used any patients with hypoxia requiring cricothyrotomy. That would have given more power to your study as you would have both cases and controls.

Reviewer #2: 1. The title is misleading

2. "It is unclear at this time how airway management via cricothyrotomy affects cerebral blood flow and hemodynamics. In this study we sought to determine the change in CBF during the palpation of neck landmarks in preparation for cricothyrotomy."

The study design enable us to see whether palpation evokes any CBF change in healthy probands and nothing more

3. Discussion

Our study has several limitations.... We do not know how the presence of a TBI would have altered our findings.

The basic data (vital data as BP CBF) of patients undergoing a cricothyrotomy are completely different than a healthy non stressed proband. You can not compare it, the only statement you can make is that CBF dos not change during palpation in healthy proband.

Conclusion

This preliminary study suggests the palpation stage is not a potential target for interventions to avoid ICP spikes during cricothyrotomy.

This statement is not correct.

6. PLOS authors have the option to publish the peer review history of their article (what does this mean?). If published, this will include your full peer review and any attached files.

Reviewer #1: No

Reviewer #2: No

---

## [Decision Letter · Decision Letter 1]

6 Jul 2020

Cerebral Blood Flow Changes During Palpation of External Airway Structures in Healthy Volunteers

PONE-D-20-03670R1

Dear Dr. Schauer,

We’re pleased to inform you that your manuscript has been judged scientifically suitable for publication and will be formally accepted for publication once it meets all outstanding technical requirements.

Kind regards,

Alon Harris

Academic Editor

PLOS ONE

Reviewers' comments:

Reviewer's Responses to Questions

**Comments to the Author**

1. If the authors have adequately addressed your comments raised in a previous round of review and you feel that this manuscript is now acceptable for publication, you may indicate that here to bypass the “Comments to the Author” section, enter your conflict of interest statement in the “Confidential to Editor” section, and submit your "Accept" recommendation.

Reviewer #1: All comments have been addressed

Reviewer #2: All comments have been addressed

2. Is the manuscript technically sound, and do the data support the conclusions?

Reviewer #1: Partly

Reviewer #2: Yes

3. Has the statistical analysis been performed appropriately and rigorously? 

Reviewer #1: Yes

Reviewer #2: Yes

4. Have the authors made all data underlying the findings in their manuscript fully available?

Reviewer #1: Yes

Reviewer #2: Yes

5. Is the manuscript presented in an intelligible fashion and written in standard English?

Reviewer #1: Yes

Reviewer #2: Yes

6. Review Comments to the Author

Reviewer #1: The authors have added a sentence on why the study was done only on healthy volunteers which was a concern that was raised during the previous review.

Reviewer #2: The authors addressed all the remarks. The name of the study as well as the discussion and conclusions were changed as suggested.

7. PLOS authors have the option to publish the peer review history of their article (what does this mean?). If published, this will include your full peer review and any attached files.

Reviewer #1: No

Reviewer #2: No

---

## [Editor Report · Acceptance letter]

15 Jul 2020

PONE-D-20-03670R1 

Cerebral Blood Flow Changes During Palpation of External Airway Structures in Healthy Volunteers 

Dear Dr. Schauer:

I'm pleased to inform you that your manuscript has been deemed suitable for publication in PLOS ONE. Congratulations! Your manuscript is now with our production department. 

Kind regards, 

on behalf of

Dr. Alon Harris 

Academic Editor

PLOS ONE